# Safety, Tolerability, and Pharmacokinetics of β-Cryptoxanthin Supplementation in Healthy Women: A Double-Blind, Randomized, Placebo-Controlled Clinical Trial

**DOI:** 10.3390/nu15102325

**Published:** 2023-05-16

**Authors:** Karen M. L. Tan, Jolene Chee, Kezlyn L. M. Lim, Maisie Ng, Min Gong, Jia Xu, Felicia Tin, Padmapriya Natarajan, Bee Lan Lee, Choon Nam Ong, Mya Thway Tint, Michelle Z. L. Kee, Falk Müller-Riemenschneider, Peter D. Gluckman, Michael J. Meaney, Mukkesh Kumar, Neerja Karnani, Johan G. Eriksson, Bindu Nandanan, Adrian Wyss, David Cameron-Smith

**Affiliations:** 1Singapore Institute for Clinical Sciences, Agency for Science Technology and Research Singapore, Singapore 117609, Singapore; jolene_chee@sics.a-star.edu.sg (J.C.); maisie_ng@sics.a-star.edu.sg (M.N.); gong_min@sics.a-star.edu.sg (M.G.); xu_jia@sics.a-star.edu.sg (J.X.); felicia_tin@sics.a-star.edu.sg (F.T.); mya_thway_tint@sics.a-star.edu.sg (M.T.T.); michelle_kee@sics.a-star.edu.sg (M.Z.L.K.); pd.gluckman@auckland.ac.nz (P.D.G.); michael.meaney@mcgill.ca (M.J.M.); mukkesh_kumar@sics.a-star.edu.sg (M.K.); neerja_karnani@sics.a-star.edu.sg (N.K.); obgjge@nus.edu.sg (J.G.E.); david.cameronsmith@newcastle.edu.au (D.C.-S.); 2Department of Laboratory Medicine, National University Hospital, Singapore 119074, Singapore; 3Bioinformatics Institute, Agency for Science Technology and Research Singapore, Singapore 138671, Singapore; 4Department of Obstetrics and Gynaecology, Yong Loo Lin School of Medicine, National University of Singapore, Singapore 117597, Singapore; obgnp@nus.edu.sg; 5Saw Swee Hock School of Public Health, National University of Singapore, Singapore 117549, Singaporeephocn@nus.edu.sg (C.N.O.); falk.m-r@nus.edu.sg (F.M.-R.); 6Human Potential Translational Research Program, Yong Loo Lin School of Medicine, National University of Singapore, Singapore 117597, Singapore; 7Digital Health Centre, Berlin Institute of Health, Charité—Universitätsmedizin Berlin, 10179 Berlin, Germany; 8Liggins Institute, University of Auckland, Auckland 1023, New Zealand; 9Douglas Mental Health University Institute, McGill University, Montreal, QC H4H 1R3, Canada; 10Department of Biochemistry, Yong Loo Lin School of Medicine, National University of Singapore, Singapore 117597, Singapore; 11Department of General Practice and Primary Health Care, University of Helsinki, 00100 Helsinki, Finland; 12Folkhälsan Research Center, 00250 Helsinki, Finland; 13DSM Nutritional Products Ltd., 4001 Basel, Switzerland; bindu.nandanan@dsm.com (B.N.); adrian.wyss@dsm.com (A.W.); 14School of Environmental and Life Sciences, University of Newcastle, Callaghan, NSW 2308, Australia

**Keywords:** carotenoids, Asian women, gene expression, mood, physical activity, fecal microbiome

## Abstract

Background: β-cryptoxanthin is a dietary carotenoid for which there have been few studies on the safety and pharmacokinetics following daily oral supplementation. Methods: 90 healthy Asian women between 21 and 35 years were randomized into three groups: 3 and 6 mg/day oral β-cryptoxanthin, and placebo. At 2, 4, and 8 weeks of supplementation, plasma carotenoid levels were measured. The effects of β-cryptoxanthin on blood retinoid-dependent gene expression, mood, physical activity and sleep, metabolic parameters, and fecal microbial composition were investigated. Results: β-cryptoxanthin supplementation for 8 weeks (3 and 6 mg/day) was found to be safe and well tolerated. Plasma β-cryptoxanthin concentration was significantly higher in the 6 mg/day group (9.0 ± 4.1 µmol/L) compared to 3 mg/day group (6.0 ± 2.6 µmol/L) (*p* < 0.03), and placebo (0.4 ± 0.1 µmol/L) (*p* < 0.001) after 8 weeks. Plasma all-trans retinol, α-cryptoxanthin, α-carotene, β-carotene, lycopene, lutein, and zeaxanthin levels were not significantly changed. No effects were found on blood retinol-dependent gene expression, mood, physical activity and sleep, metabolic parameters, and fecal microbial composition. Conclusions: Oral β-cryptoxanthin supplementation over 8 weeks lead to high plasma concentrations of β-cryptoxanthin, with no impact on other carotenoids, and was well tolerated in healthy women.

## 1. Introduction

β-cryptoxanthin is an asymmetric xanthophyll carotenoid present in high amounts in many citrus fruit species [1,2]. β-cryptoxanthin may exert beneficial actions that positively impact health with reports of significant inverse correlations between plasma concentrations and disease outcomes, including lung cancer [3], metabolic syndrome [4], blood pressure [5,6], type 2 diabetes mellitus [7,8], non-alcoholic fatty liver disease (NAFLD) [9], osteoporosis risk [10], and depressive symptoms [11,12]. Interestingly, there is also reported evidence of a possible sex-specific anti-stress correlation in females [13,14]. Further analysis in a Singapore maternal and child longitudinal cohort study, Growing Up in Singapore Towards healthy Outcomes (GUSTO), have reported that higher maternal plasma concentrations of β-cryptoxanthin were associated with better offspring cognitive and motor development, at 2 years of age [15,16]. These correlative associations suggest β-cryptoxanthin may exert both neural and metabolic impacts that could provide significant health benefits across the lifespan, including potentially during pregnancy.

β-cryptoxanthin is a member of the carotenoid family and is one of the six most prominent plasma carotenoids. However, there is limited evidence of functionality that can be attributed solely to β-cryptoxanthin, as opposed to either the combined or specific functionality of the other carotenoid species. As with other pro-vitamin A members of the carotenoid family, β-cryptoxanthin can be converted to retinol (vitamin A) via the enzymes β-carotene oxygenases 1 and 2 [1,2]. On this basis, the US Institute of Medicine (IOM) has proposed that 24 μg of β-cryptoxanthin has a vitamin A activity corresponding to 1 μg of retinol [17]. Yet, there is limited clinical data examining in vivo conversion and impact on Vitamin A dependent mechanisms with quantified β-cryptoxanthin intake.

The available human intervention trials examining the effect of β-cryptoxanthin supplementation on a variety of outcomes have reported oral doses ranging from 0.75 mg/day to 6 mg/day [14,18,19,20,21], with no major adverse reactions [18,20]. To date, only one study has involved supplemental ingestion of a β-cryptoxanthin in a purified form [18], whilst the remaining studies have used beverages or concentrates of tangerine enriched in β-cryptoxanthin [14,19,20,21]. These studies have reported on a range of endpoints, including the combined effect of β-cryptoxanthin supplements with a hypocaloric diet for 12 weeks on circulating liver enzyme status in NAFLD patients [18]. Further studies examining the actions of a Satsuma mandarin juice extract, containing β-cryptoxanthin, have demonstrated altered bone turnover markers in menopausal women [19]. Extending this analysis on bone, a milk-based beverage containing phytosterols and β-cryptoxanthin has been shown to result in significant decreases in total cholesterol, HDL-cholesterol, LDL-cholesterol, as well as alter bone turnover markers in postmenopausal women [20]. Further studies have examined possible metabolic actions with an increase in plasma adiponectin concentrations with an enriched-β-cryptoxanthin supplementation in obese, postmenopausal women [21]. Only one study has examined potential neurological actions, with Unno et al. demonstrating a reduction in salivary α-amylase activity after stressful activity on consuming a β-cryptoxanthin-rich orange drink [14]. Of these studies, three studied the effects of β-cryptoxanthin supplementation alone [14,19,21]. Yamaguchi et al. demonstrated stimulatory effects on bone formation and inhibitory effects on bone resorption of a β-cryptoxanthin-containing Satsuma mandarin juice extract at up to 6 mg/day, resulting in a plasma concentration of up to 7.2 µmol/L [19], while Iwamoto et al. showed increased adiponectin concentrations with a β-cryptoxanthin-containing beverage at up to 4.7 mg/day, resulting in a plasma concentration of up to 2.08 µmol/L [21]. These are all relatively small studies (*n* < 40 in each group), and the beneficial outcomes are not replicated as these studies focused on different outcomes.

It has been established that an increased intake of β-cryptoxanthin-containing foods elevates the plasma concentration of β-cryptoxanthin [19,22,23]. Yet, the specificity of β-cryptoxanthin supplementation in increasing only plasma β-cryptoxanthin, or whether there is in vivo conversation and potential change in retinol concentrations has not yet been examined. Given the complexity of reported outcomes, the primary objective of this study was to evaluate the pharmacokinetics of a pure oily form of β-cryptoxanthin (15% FS), ingested at a dose of 3 mg/day and 6 mg/day in women of reproductive age for 8 weeks. The dosage for this study was chosen to be comparable with previous supplementation of partially purified and concentrated Satsuma mandarin extracts (3.3–6.0 mg/day) [14,18,19,21]. The duration of this study was chosen to be 8 weeks, which is comparable with previous supplementation of β-cryptoxanthin (3 to 12 weeks) [18,19,20,21]. For this study, circulating concentrations of major carotenoid species were measured by a high-performance liquid chromatographic (HPLC) method, as previously described [24]. The secondary objective was to examine the safety and tolerability of this β-cryptoxanthin supplementation. We looked at the pro-vitamin A activity on the gene transcriptional level [1,2], with the examination of retinoic acid-dependent gene expression in circulating mononuclear cell populations [25,26]. Furthermore, carotenoids have been recently described to modify gut microbiome composition [27], hence an additional objective in this study was to determine if supplementation impacts on gut microbiome composition. Given the previous reports of sex-dependent action on mood, additional secondary outcome measures of this intervention included measures of mood, physical activity, and sleep.

## 2. Materials and Methods

### 2.1. Participants

Healthy women volunteers from Singapore, between the ages of 21 and 35, were screened to determine eligibility. The inclusion criteria included body mass index (BMI) between 18.5 to 27.5 kg/m^2^, not pregnant or breastfeeding, not planning to conceive within 3 months from enrolment, and able to comply with study instructions. Individuals were excluded if they had a current diagnosis or history of psychiatric disorders or chronic diseases, were participating in another simultaneous clinical study, were adhering to ongoing weight reduction diet or programs, were using any medication or supplementation with carotenoids, had a history of drug abuse, were current smokers or had excessive alcohol intake (>4 standard drinks per day), or had a known allergy or sensitivity to β-cryptoxanthin or corn oil. All participants voluntarily consented to participate and provided written informed consent. The study was conducted according to the guidelines laid down in the Declaration of Helsinki. Ethical approval was obtained from the National Healthcare Group Domain Specific Review Board (NHG DSRB reference 2021/00455) and the study is registered at ClinicalTrials.gov (NCT05046457).

### 2.2. Study Design

This study was a three-arm, double-blinded randomized controlled trial (RCT). Ninety study participants were randomly assigned to one of three treatment arms: (1) 3 mg/day β-cryptoxanthin for 8 weeks, (2) 6 mg/day β-cryptoxanthin for 8 weeks, and (3) placebo. β-cryptoxanthin (3 mg) formulated in 15% corn oil with 1% dl-alpha-tocopherol were packaged in soft gel capsules and was provided as one 3 mg capsule and one placebo, two 3 mg capsules and 2 placebo capsules, respectively. The capsules were manufactured by DSM Nutritional Products. The placebo consisted of identical soft gel capsules without β-cryptoxanthin. A sample size of 90 subjects, accounting for an expected drop-out rate of approximately 15%, has 80% power to detect a difference of 0.8 standard deviations on β-cryptoxanthin concentration with alpha = 0.05. The randomization codes were generated using “blockrand” R package [28] and allocated based on recruitment sequence by an unblinded statistician. A block size of 9 was used to ensure balance in sample size across the three arms. The participants attended four visits (week 0, week 2, week 4, and week 8) (Figure 1). They were required to fast overnight before each visit and vital signs were measured and blood was collected by blinded study staff. On the first and last visits (week 0 and week 8), the participants answered questionnaires on diet, health, mood, and sleep, body fat analysis was performed using bioelectrical impedance analysis (BIA) [29], and fecal samples were collected. The participants consumed 2 capsules daily for 8 weeks and compliance with the supplement and adverse events assessment were recorded at every follow-up visit by blinded study staff. Blinded parties included the investigators, sponsors, medical monitors, study site staff, staff involved in data collection and analysis, and subjects. A serious or severe adverse event is defined as an adverse event that is medically significant and life threatening or disabling, limiting activities of daily living, or requiring hospitalization or urgent intervention.

### 2.3. Carotenoid Intake

A modified food frequency questionnaire (FFQ) adapted from the Harvard-Willet FFQ [30] was designed to capture carotenoid intake from dietary sources. The modified version includes common locally consumed foods in Singapore that are also high in carotenoids [31].

### 2.4. Biochemical Analyses

Plasma carotenoids (α-cryptoxanthin, β-cryptoxanthin, α-carotene, β-carotene, lycopene, lutein, and zeaxanthin) and all-trans retinol, were measured using a HPLC method [24] from blood collected at all visits. We converted the concentrations of α-cryptoxanthin and β-cryptoxanthin from µg/mL to µmol/L by multiplying by 1.809, for α-carotene, β-carotene, and lycopene by 1.863, for lutein and zeaxanthin by 1.758 and for all-trans retinol by 3.491. Glucose, insulin, triglycerides, total cholesterol, HDL-cholesterol, LDL-cholesterol, creatinine, alanine transaminase, aspartate transaminase, and full blood count were measured in a College of American Pathologist-accredited clinical laboratory at the National University Hospital of Singapore at visits 1 (week 0) and 4 (week 8).

### 2.5. Gene Expression Analyses

Peripheral blood was collected into BD Vacutainer^®^ CPT™ Mononuclear Cell Preparation Tubes with sodium citrate and Ficoll (Becton Dickinson, Franklin Lakes, NJ, USA) and processed according to protocol. The mononuclear cell layer was rinsed with phosphate-buffered saline and stored at −80 °C in Trizol. RNA extraction was performed using miRNeasy Mini kit (Qiagen, Hilden, Germany) and the RNA integrity number (RIN) was ≥8.0 for all the RNAs. Reverse transcription was performed using Applied Biosystems™ High-Capacity cDNA Reverse Transcription Kit (ThermoFisher Scientific, Waltham, MA, USA). Real-time quantitative polymerase chain reaction (qPCR) was performed on the cDNA, using custom-designed primers to 6 genes involved in retinol signaling (retinoic receptor alpha (NM_000964) (*RARA*), retinoid X receptor alpha (NM_002957) (*RXRA*), retinoid X receptor beta (NM_021976) (*RXRB*), basic helix loop helix family member e40 (NM_003670) (*BHLHE40*), peroxisome proliferator-activated receptor delta (NM_006238) (*PPARD*), and aldehyde dehydrogenase family member A1 (NM_000689) (*ALDH1A1*)) and the human β-actin gene (NM_001101) (*ACTB*), and the Power SYBR Green Master Mix (ThermoFisher Scientific, Waltham, MA, USA) on Applied Biosystems^®^ QuantStudio™ 6 Flex Real-Time PCR System (ThermoFisher Scientific, Waltham, MA, USA). The primer sequences are shown in Appendix A. Each cDNA sample was run in duplicates for qPCR and the Cq values between duplicates did not differ by more than 0.5. Normalization was performed by calculating the mean Cq of the gene of interest—the mean Cq of ACTB (ΔCq). The difference in normalized mean Cq values was calculated for visit 4 (week 8)–visit 1 (week 0) (ΔΔCq). Fold difference in gene expression was calculated as 2^−ΔΔCq^.

### 2.6. Microbiome Analyses

Fecal samples were collected and homogenized in OMNIgene GUT OM-200 tubes (DNA Genotek, ON, Canada) and transported to the laboratory at 4 °C. The samples were vortexed and aliquoted for storage at −80 °C. Whole genomic DNA was extracted using the Promega Fecal extraction kit with beads beating, and agarose gel electrophoresis was performed as quality control for the DNA samples. The DNA samples were subject to PCR amplification for the 16S rRNA gene V3–V4 region. The amplicons were purified and subject to library preparation and quantification followed by paired-end 250bp sequencing (100K tags per sample) on the NovaSeq (Illumina, San Diego, CA, USA). DADA2 [32] was used to denoise the sequences to obtain amplicon sequence variants (ASVs) [33]. ASVs with less than 5 reads were filtered [34]. For the obtained ASVs, the representative sequence of each ASV was annotated to obtain the corresponding species information and the abundance distribution based on the species. By applying QIIME2’s classify-sklearn algorithm [35,36], a pre-trained Naive Bayes classifier against SILVA SSU Ref database 138.1 was used for species annotation of each ASV. The phylogenetic tree with the representative sequence of each ASV was constructed with MUSCLE 3.8.31 [37,38]. All the samples were standardized by rarefaction to a sequencing depth of 34,262 reads per sample, which corresponds to the lowest number of reads retained in any sample after data denoising and quality control. Alpha-diversity measurements (Shannon entropy [39] and Pielou’s evenness [40]) were calculated for each sample. Beta-diversity analysis was performed based on the weighted UniFrac [41,42] and Bray–Curtis [43] distances with the abundance of ASVs in each sample [36]. An Anosim test based on both distance matrices was performed to evaluate whether the variation among groups was significantly larger than the variation within groups. Taxon-based analysis was performed at phylum, class, order, family, genus, and species levels the by using the Mann–Whitney test.

### 2.7. Mood and Mental Well-Being

Quality of life was measured using the General Health Questionnaire (GHQ12) [44], using the Likert scoring method (0–1–2–3) [45], and the Quality of Life Enjoyment and Satisfaction Questionnaire (QLES) using the percentage maximum score transformed from the total score [46]. GHQ12 consists of 12 items, each assessing the severity of a mental problem over the past few weeks using a 4-point scale (from 0 to 3). The score was used to generate a total score ranging from 0 to 36, with higher scores indicating worse conditions. Life satisfaction measured using Satisfaction With Life Scale (SWLS) total scores [47], where 31–35 indicates Extremely satisfied, 26–30 Satisfied, 21–25 Slightly satisfied, 20 Neutral, 15–19 Slightly dissatisfied, 10–14 Dissatisfied and 5–9 Extremely dissatisfied. Mood and mental well-being were measured using the State Trait Anxiety Inventory (STAI) [48], Depression Anxiety Stress Scale (DASS) [49] and Perceived Stress Scale (PSS) [50] self-assessment questionnaires at visits 1 (week 0) and 4 (week 8). The STAI consists of two 20-item subscales (State and Trait anxiety) scored 0–4 to assess anxiety levels; the higher score indicating increased anxiety. The DASS is a 42-item self-report scale to measure the emotional states of depression, anxiety, and stress with 14 questions for each state scored on a 4-point scale: the higher score indicating increased depression, anxiety, and stress. Normal scores are 0–9 for depression, 0–7 for anxiety and 0–14 for stress [51]. The PSS is a 10-item questionnaire to measure levels of perceived stress scored on a 5-point scale: the higher the total score the higher level of perceived stress.

### 2.8. Physical Activity and Sleep

ActiGraph GT3X+ (Actigraph Inc., Pensacola, FL, USA), a triaxial accelerometer, was used to collect movement behavior data. The participants were required to wear the accelerometers for 8–9 days on their non-dominant wrist before the first (week 0) and last (week 8) visits, allowing for 7 complete days of continuous, 24 h data capture. Participants were included if they recorded minimum of 16 h of valid wear time per day for at least two valid weekdays and one weekend accelerometer data. Data were converted using the ActiLife software (version 6.13) and processed using the GGIR package (version 2.0). Night sleep duration was calculated using GGIR default algorithm and non-sleep time was classified into inactivity and light-intensity, moderate-intensity, and vigorous-intensity physical activity (<25, 25 to <100, 100 to 430 and ≥430 milligravity (mg), respectively, 1 mg = 0.00981 m.s^−2^) using prediction equations provided by Hildebrand et al. [52,53]. The Pittsburgh Sleep Quality Index (PSQI) was used to estimate sleep quality [54]. It contains 19 items that generate 7 subcomponents scores (i.e., subjective sleep quality, sleep latency, sleep duration, habitual sleep efficiency, sleep disturbances, sleep medication, and daytime functioning) on a 0–3 scale and a summed global score ranging from 0 to 21; higher scores represent poorer subjective sleep quality. Subjective poor sleep quality was defined as having a global score > 5.

### 2.9. Statistical Analyses

Statistical analyses were performed using Python (Python version 3.8.8 or higher), Stata (StataCorp LP, College Station, TX, USA; version 15 or higher), or R (The R Foundation for Statistical Computing Platform; version 4.0.3 or higher). For the primary outcome, to determine if there is a significant difference in the median change in β-cryptoxanthin levels from week 0 to week 2, week 0 to week 4 and week 0 to week 8 between the treatment groups, a Kruskal–Wallis test was conducted. To further find out the extent of differences in the median change in β-cryptoxanthin levels between visits between the groups, the Mann–Whitney U test was conducted. For secondary outcomes, continuous endpoints that are approximately normally distributed or can be transformed to be approximately normally distributed were analyzed using repeated measures of one-way analysis of variance (ANOVA) to test for differences between the means of the control and intervention groups. Continuous endpoints that are not normally distributed were analyzed using a Kruskal–Wallis test to test for a difference in the median between the control and intervention groups. Statistical tests were conducted two-sided, and a *p*-value of less than 0.05 was considered statistically significant.

## 3. Results

Overall, 89% of all randomized participants (*n* = 80) completed the study (Figure 2). The mean compliance rate was 95%. The baseline study demographics, carotenoid intakes, and plasma concentrations are shown in Table 1. There were no significant differences at baseline between placebo and treatment dose groups, except for reported retinol intake, which was highest in the 6 mg/day β-cryptoxanthin treatment group. However, the baseline plasma all-trans-retinol concentrations were not significantly different between the placebo and treatment dose groups. Baseline β-cryptoxanthin daily intake and plasma β-cryptoxanthin concentrations were not different between the placebo and treatment dose groups.

### 3.1. Safety and Tolerability

No serious or severe adverse events or deaths were observed. All reported adverse events are summarized in Table 2. The most common adverse event reported was COVID-19 infection, followed by palmar yellowness or carotenemia, a yellowing of the skin due to the accumulation of carotenoids which was considered to be harmless [46]. In addition to carotenemia, no other adverse events were determined to be study related. No gastrointestinal discomfort or headache was reported. Table 3 shows the vital signs and laboratory parameters at baseline and after the 8 weeks supplementation period. No clinically significant changes in vital signs or laboratory parameters that were considered to be related to β-cryptoxanthin were observed after 8 weeks of supplementation of up to 6 mg/day.

### 3.2. Pharmacokinetics of Plasma Carotenoids

The mean concentration-time profiles of the plasma carotenoids over 8 weeks are shown in Figure 3. The mean plasma concentration of β-cryptoxanthin increased with time in a dose-dependent manner in the β-cryptoxanthin treated groups but not in the placebo group. Following 2 weeks of daily consumption of β-cryptoxanthin, the plasma β-cryptoxanthin concentration increased to 3.5 ± 1.6 µmol/L from a baseline of 0.6 ± 0.5 µmol/L in the 3 mg/day group and to 4.2 ± 1.2 µmol/L from a baseline of 0.4 ± 0.2 µmol/L in the 6 mg/day group. At 2 weeks, the concentration of β-cryptoxanthin in the 3 and 6 mg/day groups were significantly higher compared to the control (Figure 3A). After 8 weeks of daily consumption of β-cryptoxanthin, the plasma β-cryptoxanthin concentration further increased to 6.0 ± 2.6 µmol/L in the 3 mg/day group and to 9.0 ± 4.1 µmol/L in the 6 mg/day group. At 4 and 8 weeks, the concentration of β-cryptoxanthin in the 3 and 6 mg/day groups were significantly higher compared to the control (*p* < 0.001). Further, the concentration of β-cryptoxanthin in the 6 mg/day group was significantly higher compared to that in the 3 mg/day group (*p* = 0.03) (Figure 3A). The mean plasma concentrations of the other carotenoids (α-cryptoxanthin, β-carotene, lycopene, lutein, and zeaxanthin) and all-trans retinol did not change with time in all three groups (Figure 3B–F,H), except for α-carotene, which increased slightly over time in the placebo as well as the treatment groups at 4 and 8 weeks (0.26 ± 0.43 µmol/L at week 8 vs. 0.09 ± 0.04 µmol/L at baseline for placebo) (Figure 3G).

### 3.3. Gene Expression

Figure 4 shows the fold changes of gene expression of the 6 retinol-dependent genes in peripheral blood mononuclear cells (PBMCs) from baseline after the 8 weeks of supplementation period. Overall, 8 weeks of supplementation of β-cryptoxanthin up to 6 mg/day did not significantly change the gene expression of *RARA*, *RXRA*, *RXRB*, *BHLHE40*, *PPARD*, and *ALDH1A1* in PBMCs. The differences in fold changes of gene expression after 8 weeks of supplementation were not statistically significant in all the three treatment groups.

### 3.4. Gut Microbiome

After quality filtration, a total of 6,152,753,266 high-quality reads were obtained from 160 samples with an average of 5,786,269 ± 38,454,708 reads per sample. By using DADA2 denoising, 17,915 ASVs were obtained for the downstream microbiome analysis. Shannon entropy and Pielou’s evenness were calculated to evaluate the microbial diversity and evenness in all the samples. There was no significant difference in microbial community diversity between the samples of the three different treatment groups before and after supplementation (Figure 5).

As indicated by the principal coordinate analysis (PCoA) based on Bray–Curtis and Weighted UniFrac distances (Figure 6), samples of all the three different treatment groups clustered together without a distinct classification of samples before and after supplementation. Anosim test based on the Bray–Curtis and Weighted Unifrac distances showed that there was no significant difference in the microbial composition between the samples of the three different treatment groups before and after supplementation. Taxon-based analysis was performed at phylum, class, order, family, genus, and species levels. No phylum, class, or order was significantly different between before and after supplementation in the three treatment groups after correcting for multiple comparisons.

Overall, the results indicated that β-cryptoxanthin consumption of up to 6 mg/day had no significant effect on microbial composition compared with placebo and did not alter the relative abundance of individual microbial species compared with baseline.

### 3.5. Mood, Physical Activity, and Sleep

Table 4 shows the data on study outcomes related to mood, physical activity, and sleep at baseline and after the 8 weeks supplementation period. No significant alterations in quality-of-life scores, depression, stress and anxiety scores, and physical activity and sleep patterns were observed.

## 4. Discussion

This is the first RCT to examine the effects of a pure encapsulated form of β-cryptoxanthin in healthy women. Multiple doses of β-cryptoxanthin of up to 6 mg/day for 8 weeks were well tolerated by the women in this study. No serious adverse events were observed, and none of the reported adverse events were determined to be related to β-cryptoxanthin except carotenemia, a benign yellowing of the skin [55]. Carotenemia is a harmless condition due to the accumulation of β-cryptoxanthin in the skin but may be mistakenly seen as jaundice. β-cryptoxanthin was not associated with any clinically significant findings in vital signs or clinical laboratory parameters, including the liver enzymes aspartate transaminase and alanine transaminase. This suggests that β-cryptoxanthin is safe and has minimal risk as a nutritional supplement.

The mean β-cryptoxanthin concentration at baseline was 0.44 ± 1.2 µmol/L, which is similar to that observed in mothers at the time of delivery in the GUSTO cohort [6]. The pharmacokinetic properties of β-cryptoxanthin observed in this study are comparable to those reported with enriched fruit-derived supplements [19,22,23]. We observed an increase in plasma β-cryptoxanthin concentration from 0.4 ± 0.2 µmol/L to 9.0 ± 4.1 µmol/L (22.5-fold increase) in the women after 8 weeks of daily supplementation with β-cryptoxanthin at 6 mg/day. Similarly, Yamaguchi [19] reported that with a daily intake of a concentrated Satsuma mandarin juice containing β-cryptoxanthin at 6 mg/day, serum β-cryptoxanthin concentrations increased from 0.4 µmol/L to 5.9 µmol/L (14.75-fold increase) after 28 days in healthy volunteers. In contrast, Haidari et al. [18] reported an increase in serum β-cryptoxanthin concentration from 0.29 µmol/L (0.15 µg/mL) at baseline to 0.51 µmol/L (0.26 µg/mL) (1.8-fold increase) after 12 weeks of daily supplementation with a capsule form of β-cryptoxanthin at 6 mg/day in patients with NAFLD. These results indicate that the β-cryptoxanthin capsule used in our study likely has higher or similar bioavailability compared to the bioavailability of natural concentrated Satsuma mandarin juice and other capsule formulations [18,19,21,56], although the studies are not directly comparable as the treatment duration and study populations are different.

Given the purity of the current β-cryptoxanthin and ingestion as a supplemental dose, this study was able to examine the subsequent impact on circulating concentrations of additional carotenoid species. β-cryptoxanthin supplementation at a dose of both 3 and 6 mg increased plasma β-cryptoxanthin concentration without any corresponding alterations in the plasma concentration of other carotenoids, including α-carotene, β-carotene, lycopene, lutein, and zeaxanthin. Our study extends these insights further to examine the effects of β-cryptoxanthin supplementation on the gene expression involved in retinol signaling in PBMCs. For this, we selected six candidate genes based on their gene expression level within human blood cells in the Genotype-Tissue Expression (GTEx) database (https://gtexportal.org/home/aboutGTEx, last accessed 15 September 2022) and reported sensitivity to retinoic acid (Vitamin A). Oral supplementation with up to 6 mg/day of β-cryptoxanthin for 8 weeks did not change the gene expression of any of these retinoic acid-dependent genes. This is consistent with the finding that there was no significant change in plasma all-trans retinol concentrations and further suggests that the β-cryptoxanthin supplement may exert limited impact on retinoid-dependent gene signaling. In the context of the current study, this is an important finding in that the women recruited for this study were of childbearing age. Although those pregnant and planning to become pregnant were excluded from the study, it is possible that pregnancy initiation might have occurred during the trial. Given the relationship between high-dose vitamin A intake and fetal teratotoxicity [57,58], the possibility of β-cryptoxanthin as a fetal teratogen could not be completely ignored in both the current study and potentially in further studies examining the specificity of β-cryptoxanthin’s impacts on the health of women of childbearing age. A limitation of this study is that the retinol-dependent gene expression was examined only in PBMCs and the effect of β-cryptoxanthin on retinol-dependent gene expression in other tissues, such as the liver and neuronal tissues is still unknown. From in vivo evidence it is further known that the metabolic conversion of pro-vitamin A carotenoids to retinoids and the uptake mechanisms of these carotenoids is regulated by the intestinal transcription factor ISX, preventing retinoid synthesis after absorption [59]. In this clinical trial we could demonstrate that a high nutritional dose of a pro-vitamin A carotenoid did not lead to harmful effects generated by unbalanced retinoic acid synthesis.

To our knowledge the current study is also the first study to examine the effects of β-cryptoxanthin supplementation on the gut microbiome. No significant differences in fecal microbial composition were observed after β-cryptoxanthin daily supplementation for 8 weeks, with a dose of 3 mg/day or 6 mg/day, compared to placebo. It has been reported that dietary and plasma carotenoids are positively associated with alpha diversity of fecal microbiota [60]. However, the association may reflect an effect of high fiber or improved overall dietary quality, rather than a specific causal effect of carotenoids, on the microbiota. A limitation of this study is that the fecal microbial composition was studied after 8 weeks hence we are unable to study the long-term effects of β-cryptoxanthin on the gut microbiome.

Previous studies have reported a reduction in stress levels in female university students as measured by a reduction in salivary α-amylase activity on consuming β-cryptoxanthin in the form of concentrated Satsuma mandarin juice [13,14]. In the current study, multiple measures of stress status were conducted, including anxiety, depression, quality of life and satisfaction with life. However, β-cryptoxanthin supplementation did not significantly impact the subjective scores of any of these mood measures (Table 4). This is not surprising, as this study was not powered for the secondary objectives. In addition, we recruited a healthy female population with low subjective scores of depression, anxiety, and stress and high scores of well-being and satisfaction with life. Given the sex-specific findings from the concentrated Satsuma mandarin juice [13,14], analysis of the actions of β-cryptoxanthin supplementation in a population selected on the basis of low mood or high stress, is required.

In relation to health, higher plasma β-cryptoxanthin concentrations have previously been associated with lower blood pressure during late pregnancy [6]. In this study there was no change in systolic and diastolic blood pressure in the β-cryptoxanthin treated groups compared with the placebo (Table 3). Hence, the association observed in the GUSTO cohort may reflect a healthier lifestyle with increased fruit and vegetable consumption resulting in higher plasma β-cryptoxanthin concentrations rather than a causal effect of β-cryptoxanthin on lowering blood pressure. Carotenoids, including β-cryptoxanthin, concentrations have been shown to be inversely correlated with BMI [61] and metabolic health [4,5,62]. However, in this study, no significant changes in BMI, fat percentage, fasting glucose and insulin, cholesterol and triglyceride concentrations, and physical activity were observed in the β-cryptoxanthin treated groups compared with placebo. Similarly, Iwamoto et al. found that BMI, blood pressure, fasting glucose, cholesterol, and triglyceride concentrations were not significantly changed in obese Japanese women after 3 weeks of β-cryptoxanthin intake in the form of a concentrated Satsuma mandarin juice [21]. In this current study, which was not powered for the secondary objectives, all the women recruited were relatively young (average age 25), lean (average BMI 21 kg/m^2^) and healthy (average blood pressure 110/70 mmHg and average fasting glucose 4.7 mmol/L). This may be one of the reasons no effects of β-cryptoxanthin supplementation on metabolic health were observed. Further studies on obese women with metabolic syndrome will be required to determine whether β-cryptoxanthin supplementation has an effect on improving metabolic health.

The strengths of this study were the use of a β-cryptoxanthin formulation with a very good bioavailability and the extensive amount of data collected. One limitation is the lack of follow-up after 8 weeks of β-cryptoxanthin supplementation to observe the long-term effects. Another limitation is that the study was underpowered such that secondary outcomes could not be adequately investigated. This study involved only healthy Asian women. It would be interesting to determine whether men and other ethnic groups would respond in the same way to β-cryptoxanthin supplementation. Overall, this pharmacokinetic human clinical trial in healthy women serves well for future clinical trials where the effects of β-cryptoxanthin supplementation on metabolic or mental health in women are investigated.

## 5. Conclusions

β-cryptoxanthin ingested as a purified supplement was found to be safe and well-tolerated at doses up to 6 mg/day for 8 weeks in healthy women. The pharmacokinetic profile demonstrated that β-cryptoxanthin plasma concentrations increased in a dose-dependent manner over time following daily supplementation. No significant changes in plasma concentrations of other carotenoids, blood retinol-dependent gene expression, mood, physical activity and sleep, metabolic parameters, and fecal microbial composition were observed after 8 weeks of supplementation with β-cryptoxanthin at up to 6 mg/day in healthy women.

## Figures and Tables

**Figure 1 nutrients-15-02325-f001:**
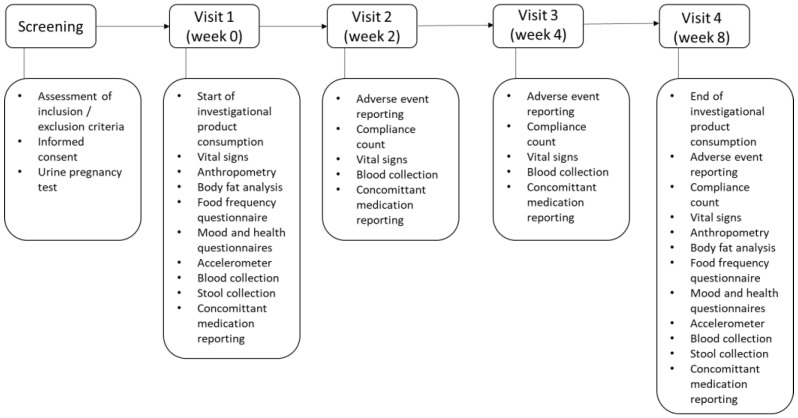
Flowchart of study visits.

**Figure 2 nutrients-15-02325-f002:**
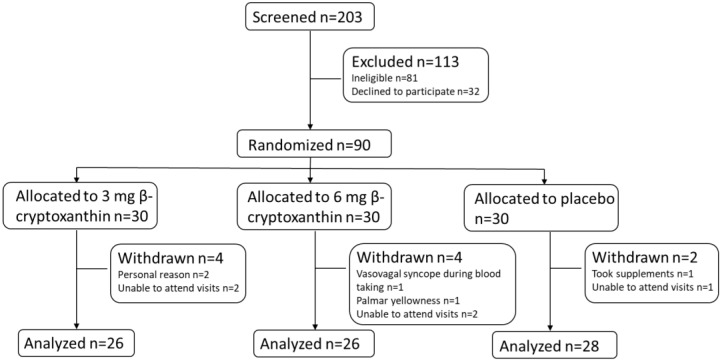
Flowchart of participants.

**Figure 3 nutrients-15-02325-f003:**
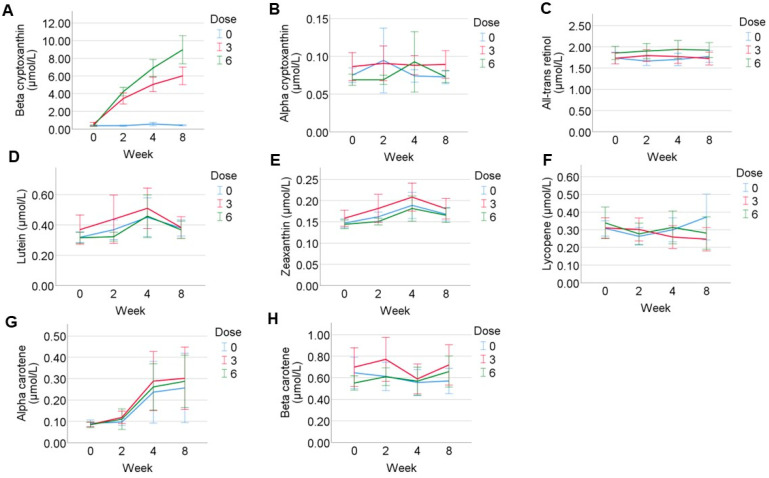
Time-course of plasma carotenoid concentrations. Mean plasma carotenoid concentrations in (µmol/L) ± standard error of the mean are shown at weeks 0, 2, 4, and 8. The placebo group is represented with blue lines, the 3 mg β-cryptoxanthin group is represented with red lines, and the 6 mg β-cryptoxanthin group is represented with green lines. (**A**) β-cryptoxanthin, (**B**) α-cryptoxanthin, (**C**) all-trans retinol, (**D**) lutein, (**E**) zeaxanthin, (**F**) lycopene, (**G**) α-carotene, and (**H**) β-carotene.

**Figure 4 nutrients-15-02325-f004:**
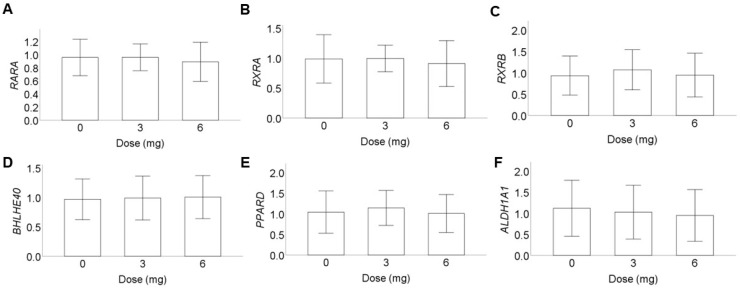
Fold changes of retinol-dependent gene expression in peripheral blood mononuclear cells from placebo and β-cryptoxanthin supplementation groups. Fold changes in gene expression from baseline to week 8 were calculated using normalized Cq values. Values are expressed as mean ± standard deviation. Dose of β-cryptoxanthin in mg per day is indicated on the *x*-axis. (**A**) *RARA*, (**B**) *RXRA*, (**C**) *RXRB*, (**D**) *BHLHE40*, (**E**) *PPARD*, and (**F**) *ALDH1A1*.

**Figure 5 nutrients-15-02325-f005:**
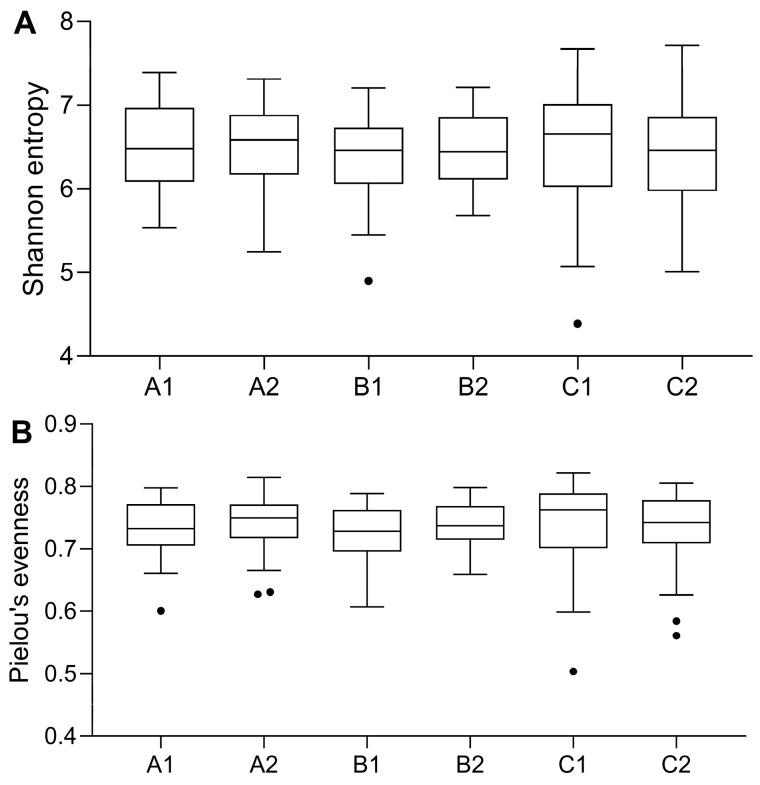
Alpha diversity of the gut microbial community in 3 different treatment groups before and after supplementation. (**A**) Shannon entropy. (**B**) Pielou’s evenness. All the data were expressed with Tukey style boxplots. The dots represent outliers. A1: 3 mg/day β-cryptoxanthin at baseline, A2: 3 mg/day β-cryptoxanthin after 8 weeks, B1: 6 mg/day β-cryptoxanthin at baseline, B2: 6 mg/day β-cryptoxanthin after 8 weeks, C1: placebo at baseline, C2: placebo after 8 weeks.

**Figure 6 nutrients-15-02325-f006:**
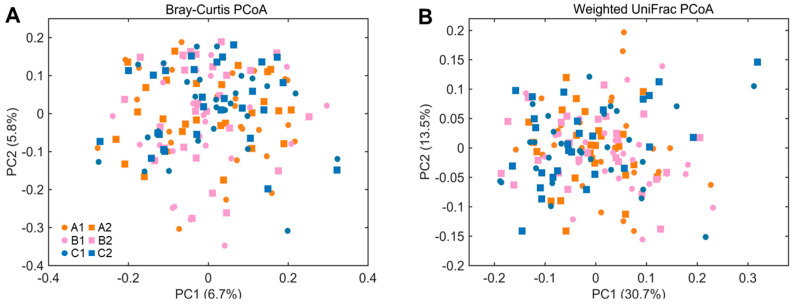
Beta diversity of the gut microbial community in 3 different treatment groups before and after supplementation. Principal coordinate analysis based on (**A**) Bray-Curtis and (**B**) Weighted UniFrac distances. A1: 3 mg/day β-cryptoxanthin at baseline, A2: 3 mg/day β-cryptoxanthin after 8 weeks, B1: 6 mg/day β-cryptoxanthin at baseline, B2: 6 mg/day β-cryptoxanthin after 8 weeks, C1: placebo at baseline, C2: placebo after 8 weeks.

**Table 1 nutrients-15-02325-t001:** Baseline study demographics and carotenoid intake and plasma concentrations.

	Placebo	3 mg β-Cryptoxanthin	6 mg β-Cryptoxanthin	*p*-Value
Participants (*n*)	28	26	26	
Age (years)	27.5 (8.5)	22.5 (7.75)	25.5 (7.5)	0.131
Height (cm)	162.01 ± 5.87	161.1 ± 4.28	161.6 ± 4.28	0.794
Weight (kg)	58.3 ± 7.68	55.9 ± 6.75	57.1 ± 6.31	0.446
BMI (kg/m^2^)	21.75 (3.70)	21.2 (3.75)	21.65 (3.56)	0.594
Ethnicity (*n*(%))				
Chinese	22 (78.6%)	25 (96.2%)	24 (92.3%)	
Malay	0 (0%)	0 (0%)	1 (3.8%)	
Indian	6 (21.4%)	0 (0%)	1 (3.8%)	
Others	0 (0%)	1 (3.8%)	0 (0%)	
Carotenoid intake				
Retinol intake (μg/day)	179 (677)	228 (5433)	1136 (5777)	0.039
β-carotene intake (μg/day)	2359 (3123)	2550 (2566)	2909 (2534)	0.790
α-carotene intake (μg/day)	243 (390)	168 (279)	316 (385)	0.272
β-cryptoxanthin intake (μg/day)	110 (119)	117 (137)	117 (99)	0.751
Lycopene intake (μg/day)	1192 (1173)	649 (915)	1406 (1526)	0.115
Lutein + zeaxanthin intake (μg/day)	1829 (1692)	2219 (2244)	1749 (1659)	0.695
Plasma carotenoid concentrations				
All-trans-retinol (μmol/L)	1.68 (0.42)	1.68 (0.45)	1.78 (0.38)	0.418
β-carotene (μmol/L)	0.54 (0.32)	0.54 (0.43)	0.56 (0.26)	0.913
α-carotene (μmol/L)	0.07 (0.06)	0.07 (0.02)	0.07 (0.04)	0.870
β-cryptoxanthin (μmol/L)	0.34 (0.18)	0.36 (0.24)	0.33 (0.33)	0.511
α-cryptoxanthin (μmol/L)	0.07 (0.02)	0.07 (0.02)	0.07 (0.02)	0.329
Lycopene (μmol/L)	0.32 (0.15)	0.30 (0.15)	0.28 (0.17)	0.961
Lutein (μmol/L)	0.30 (0.09)	0.32 (0.14)	0.32 (0.09)	0.997
Zeaxanthin (μmol/L)	0.14 (0.04)	0.16 (0.05)	0.14 (0.04)	0.657

Data are presented as median (interquartile range) or mean ± standard deviation unless otherwise indicated. *p*-values were obtained by Kruskal–Wallis test and one way analysis of variance for not normally distributed and normally distributed data, respectively.

**Table 2 nutrients-15-02325-t002:** Summary of reported adverse events.

	Placebo	3 mg β-Cryptoxanthin	6 mg β-Cryptoxanthin	Total
Any event	3	2	13	18
COVID-19	2	0	5	7
Carotenemia	0	1	5	6
Skin rashes	0	0	1	1
Urticaria	0	1	0	1
Acute upper respiratory tract infection	1	0	0	1
Urinary tract infection	0	0	1	1
Eye infection	0	0	1	1

**Table 3 nutrients-15-02325-t003:** Vital signs and laboratory parameters at baseline and after 8 weeks supplementation.

	Placebo	3 mg β-Cryptoxanthin	6 mg β-Cryptoxanthin	*p*-Value
BMI (kg/m^2^)				
Baseline	21.75 (3.70)	21.2 (3.75)	21.35 (3.56)	0.594
Week 8	22 (3.17)	21.1 (3.45)	21.35 (3.63)	0.494
*p*-value	0.896	0.957	0.898	
Fat percentage (%)				
Baseline	30.77 ± 7.02	28.36 ± 4.64	30.85 ± 5.75	0.252
Week 8	31.82 ± 7.46	28.89 ± 4.58	31.42 ± 5.39	0.102
*p*-value	0.599	0.694	0.714	
Systolic blood pressure (mmHg)				
Baseline	110.79 ± 8.88	110.46 ± 8.65	109.08 ± 9.11	0.759
Week 8	108.25 ± 10.53	109.15 ± 6.66	110.92 ± 8.97	0.539
*p*-value	0.334	0.544	0.465	
Diastolic blood pressure (mmHg)				
Baseline	72.00 ± 6.98	71.15 ± 7.92	69.81 ± 8.00	0.572
Week 8	70.64 ± 6.74	69.27 ± 7.01	70.92 ± 7.94	0.677
*p*-value	0.462	0.368	0.616	
Heart rate (bpm)				
Baseline	70.96 ± 7.82	73.88 ± 11.41	72.85 ± 8.16	0.262
Week 8	73.07 ± 8.46	72.96 ± 10.73	73.81 ± 12.88	0.905
*p*-value	0.175	0.620	0.615	
Haemoglobin (g/dL)				
Baseline	12.45 (1.30)	12.30 (1.28)	12.15 (1.55)	0.972
Week 8	12.35 (1.20)	12.15 (1.53)	12.60 (1.72)	0.889
*p*-value	0.717	0.636	0.992	
Haematocrit **(%)**				
Baseline	37.80 (3.85)	37.75 (3.22)	38.30 (4.42)	0.732
Week 8	38.00 (2.85)	37.35 (3.47)	38.85 (4.40)	0.822
*p*-value	0.447	0.545	0.957	
Creatinine (μmol/L)				
Baseline	63.5 (9.8)	61.0 (9.3)	62.0 (14.8)	0.716
Week 8	60.0 (14.0)	60.0 (7.0)	63.0 (12.5)	0.964
*p*-value	0.620	0.962	0.922	
Alanine transaminase (U/L)				
Baseline	10.5 (6.8)	11.5 (6.3)	12.0 (4.0)	0.626
Week 8	13.0 (7.0)	11.5 (4.8)	13.0 (5.5)	0.156
*p*-value	0.225	0.953	0.346	
Aspartate transaminase (U/L)				
Baseline	19.5 (5.8)	21.0 (5.3)	20.0 (3.0)	0.530
Week 8	20.0 (7.0)	19.0 (3.8)	21.0 (7.0)	0.131
*p*-value	0.560	0.165	0.282	
Glucose (mmol/L)				
Baseline	4.74 ± 0.24	4.75 ± 0.33	4.74 ± 0.35	0.991
Week 8	4.62 ± 0.36	4.72 ± 0.28	4.76 ± 0.44	0.361
*p*-value	0.184	0.792	0.862	
Insulin (mU/L)				
Baseline	4.60 (2.13)	5.15 (3.05)	5.90 (2.98)	0.217
Week 8	4.60 (3.10)	4.50 (3.28)	5.95 (3.60)	0.338
*p*-value	0.858	0.339	0.402	
Total cholesterol (mmol/L)				
Baseline	4.68 (0.91)	4.82 (1.32)	4.74 (1.22)	0.954
Week 8	4.82 (1.12)	4.72 (1.19)	4.74 (1.44)	0.897
*p*-value	0.393	0.372	0.992	
Triglycerides (mmol/L)				
Baseline	0.71 (0.32)	0.71 (0.26)	0.85 (0.45)	0.070
Week 8	0.63 (0.23)	0.74 (0.34)	0.73 (0.35)	0.208
*p*-value	0.707	0.672	0.613	
HDL-cholesterol (mmol/L)				
Baseline	1.61 (0.30)	1.61 (0.45)	1.60 (0.37)	0.735
Week 8	1.70 (0.29)	1.60 (0.34)	1.54 (0.40)	0.538
*p*-value	0.595	0.851	0.571	
LDL-cholesterol (mmol/L)				
Baseline	2.66 (0.78)	2.76 (1.10)	2.61 (0.85)	0.936
Week 8	2.72 (1.02)	2.55 (1.39)	2.74 (1.01)	0.964
*p*-value	0.449	0.660	0.276	

Data are presented as median (interquartile range) or mean ± standard deviation unless otherwise indicated. *p*-values were obtained by Kruskal–Wallis test and one way analysis of variance for not normally distributed and normally distributed data, respectively.

**Table 4 nutrients-15-02325-t004:** Mood, physical activity, and sleep at baseline and after 8 weeks supplementation.

	Placebo	3 mg β-Cryptoxanthin	6 mg β-Cryptoxanthin	*p*-Value
General health questionnaire (GHQ) scores				
Baseline	11.0 (5.0)	10.5 (4.5)	10.0 (5.0)	0.669
Week 8	10.0 (5.0)	10.0 (6.5)	10.0 (5.0)	0.529
*p*-value	0.383	0.582	0.613	
Quality of Life Enjoyment and Satisfaction Questionnaire (QLES) scores				
Baseline	68.6 (14.6)	70.0 (12.9)	71.7 (13.4)	0.771
Week 8	71.7 (12.1)	74.2 (11.2)	71.7 (15.7)	0.669
*p*-value	0.229	0.073	0.921	
Satisfaction With Life Scale (SWLS) scores				
Baseline	22.5 (12.0)	25.0 (10.0)	25.0 (6.8)	0.411
Week 8	24.5 (9.3)	23.5 (6.0)	26.5 (7.0)	0.308
*p*-value	0.555	0.499	0.596	
State Trait Anxiety Inventory (STAI) Trait scores				
Baseline	43.5 (10.5)	41.0 (8.8)	41.0 (12.8)	0.519
Week 8	38.0 (12.3)	40.0 (8.8)	42.0 (9.0)	0.339
*p*-value	0.082	0.749	0.702	
State Trait Anxiety Inventory (STAI) State scores				
Baseline	39.0 (15.0)	39.0 (9.8)	34.5 (13.8)	0.449
Week 8	35.5 (14.5)	34.5 (11.5)	37.0 (12.0)	0.693
*p*-value	0.209	0.134	0.730	
Depression Anxiety Stress Scale (DASS) Depression scores				
Baseline	5.5 (11.3)	3.0 (5.8)	2.0 (6.8)	0.455
Week 8	2.0 (9.3)	2.0 (2.0)	4.0 (4.8)	0.667
*p*-value	0.181	0.650	0.208	
Depression Anxiety Stress Scale (DASS) Anxiety scores				
Baseline	4.0 (3.8)	4.0 (5.5)	3.5 (4.5)	0.928
Week 8	3.0 (4.3)	3.0 (4.0)	3.0 (4.5)	0.806
*p*-value	0.115	0.462	0.492	
Depression Anxiety Stress Scale (DASS) Stress scores				
Baseline	7.5 (7.5)	6.5 (5.8)	7.5 (8.8)	0.711
Week 8	4.5 (11.0)	6.5 (7.8)	7.0 (8.5)	0.525
*p*-value	0.259	0.770	0.463	
Perceived Stress Scale (PSS) scores				
Baseline	18.0 (5.0)	16.5 (6.5)	16.5 (9.8)	0.777
Week 8	14.0 (9.3)	15.5 (7.0)	15.5 (5.0)	0.970
*p*-value	0.146	0.640	0.617	
Inactivity (sedentary) (min/day)				
Baseline	657.8 (123.6)	650.1 (132.2)	689.5 (107.3)	0.059
Week 8	625.0 (112.4)	658.6 (149.4)	610.8 (128.3)	0.919
*p*-value	0.919	0.970	0.073	
Light physical activity (min/day)				
Baseline	253.8 (85.6)	278.2 (71.5)	235.4 (80.7)	0.007
Week 8	253.2 (81.6)	237.1 (132.7)	246.1 (95.1)	0.284
*p*-value	0.755	0.536	0.589	
Moderate physical activity (min/day)				
Baseline	87.3 ± 38.3	92.5 ± 30.2	79.3 ± 31.8	0.391
Week 8	89.4 ± 28.4	91.9 ± 40.8	84.0 ± 30.5	0.716
*p*-value	0.827	0.957	0.605	
Vigorous physical activity (min/day)				
Baseline	2.9 (3.9)	3.6 (11.3)	3.0 (4.7)	0.158
Week 8	2.9 (3.8)	4.7 (10.3)	3.1 (5.4)	0.522
*p*-value	0.661	0.398	0.268	
Sleep (min/day)				
Baseline	433.6 (79.8)	418.6 (59.2)	431.5 (52.5)	0.279
Week 8	440.8 (69.5)	417.4 (106.4)	432.9 (100.7)	0.311
*p*-value	0.985	0.636	0.160	
Pittsburgh Sleep Quality Index (PSQI) scores				
Baseline	6.0 (3.0)	5.0 (3.0)	5.0 (2.0)	0.249
Week 8	6.0 (4.0)	5.5 (3.0)	5.0 (3.0)	0.499
*p*-value	0.980	0.201	0.778	

Data are presented as median (interquartile range) or mean ± standard deviation unless otherwise indicated. *p*-values were obtained by Kruskal–Wallis test and one way analysis of variance for not normally distributed and normally distributed data, respectively.

## Data Availability

The data presented in this study are available on request from the corresponding author.

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
