# Peer review of "Safety, Tolerability, and Pharmacokinetics of β-Cryptoxanthin Supplementation in Healthy Women: A Double-Blind, Randomized, Placebo-Controlled Clinical Trial"

_nutrients, 2023, doi:10.3390/nu15102325_

Round 1

Reviewer 1 Report

β-cryptoxanthin is rich in fruits and has showed some beneficial effects in multiple diseases according to the published studies. This manuscript reported β-cryptoxanthin supplementation for 8 weeks (3 and 6 mg/day) in 90 healthy Asia women between 21 to 35 years old doesn't have any effects on other carotenoid concentrations, retinol-dependent gene expression, microbiome, mood, physical activity, sleep, metabolic parameters, indicating the treatment safety of β-cryptoxanthin alone in healthy people. There are some questions and concerns are listed as following:

(1)  Whether β-cryptoxanthin alone has been tested in clinic for the diseases, how strong of its beneficial effects and what’s the potential of it to become a drug candidate? Please clarify and discuss.

(2)  Although people haven't reported the safety of β-cryptoxanthin alone, how potent of β-cryptoxanthi decides the importance of the work, this would also be the major concern.

(3)  Line 366-377, when compare the bioavailability, since the treatment period is totally different in the studies, it may not be reasonable to make conclusion that β-cryptoxanthin has higher bioavailability in your study. 

(4)  In table 4, please clarify the units for the numbers, if they are scores, please simply introduce the scoring method in the section of methods.

(5)  Please relate the detailed figure panels to the text for section 3.2.

Some sentences are not friendly read and suggested to be improved, for example line 27-28, line 246-252, 288 etc.

Reviewer 2 Report

Tran et al. investigated pharmacokinetics and impacts of beta-cryptoxanthin in healthy Asian women. In the study, subjects were separated into three groups, received 3 mg beta-cryptoxanthin, 6 mg beta-cryptoxanthin, or placebo respectively for eight weeks. By a series of tests regarding to mood, physical activity and sleep, metabolic parameters, as well as fecal microbiota concentration, the author found that beta-cryptoxanthin is well-tolerant in these subjects.

The introduction provides a clear background and rationale for the study, and the research question is well-defined. However, for the experimental design, is there any specific consideration for choosing 8 weeks as the timeline? The choice of 8 weeks as the duration of the study may have been based on previous studies or literature that indicated this to be a reasonable time frame to observe changes in beta-cryptoxanthin levels and its effects on health outcomes. Alternatively, it could have been based on practical considerations such as the availability of funding or participant availability. Thus, it would be helpful if the author can provide some background on this.

Regarding the figures, it would be helpful to address the formatting issues  to improve the clarity of the information presented. For example, part of the legend (dose) was cut in Figure 3C and F. The font size in label of x- and y- axis from Figure 3 and 4 is too small. Please increase the font size.

L201-203 “Alpha-diversity measurements (Shannon entropy and Pielou’s evenness) were calculated for each sample. Beta-diversity analysis was performed based on the weighted and Bray-Curtis distances with the abundance of ASVs in each sample.“

Please add reference for diversity measurement.

Reviewer 3 Report

Critique Nutrient Manuscript-2369136

Safety, tolerability and pharmacokinetics of b-cryptoxanthine supplementation in healthy women: a double blind randomized placebo-controlled clinical trial

Karen Tan, Jolene Chee, Kezlyn Lim, Maisie Ng, et al.

Overall Impression:  For many years I have been pleading with investigators to move from animal studies to clinical trials to study actions of natural/nutritional compounds.  Thus, I was pleased to see this manuscript describing a well-controlled with an extensive number of biological processes being measured.  Of course, almost nothing is perfect and I have mentioned these below.

Specific Comments:  

1.    The study involved only Asian healthy women.  It would be interesting to determine whether other ethnic populations would respond the same way to the Beta-cryptoxanthine.

2.   The study was underpowered such that, as mentioned by the authors, secondary outcomes could not be adequately investigated.

3.   I was quite impressed with the systolic blood pressures for these women and this indicates that it likely was a health population and may also reflect the life style among the Singapore population.

4.   The same comment as above also is reflected in their glucose levels

5.   Overall this “baseline” human clinical trial serves well for other clinical trials where some health problems and their potential amelioration by beta-cryptoxanthine supplementation is investigated.

Round 2

Reviewer 1 Report

1. In figure 3,  label "C" is missing ?

2. Please unify the font and font size in the figures.

None.

Author Response

Response to reviewer 1:

Comment 1: In figure 3,  label "C" is missing ?

Response 1: The label "C" is now visible in figure 3 in the revised manuscript.

Comment 2:  Please unify the font and font size in the figures.

Response 2: The fonts and font sizes are unified in the figures.